# ImageFolder: Autoregressive Image Generation with Folded Tokens

**Xiang Li**[*,1,2], **Kai Qiu**[1], **Hao Chen**[1], **Jason Kuen**[2], **Jiuxiang Gu**[2], **Bhiksha Raj**[1,3], **Zhe Lin**[2]
Carnegie Mellon University[1], Adobe Research[2], MBZUAI[3]
Project Page: ImageFolder.github.io

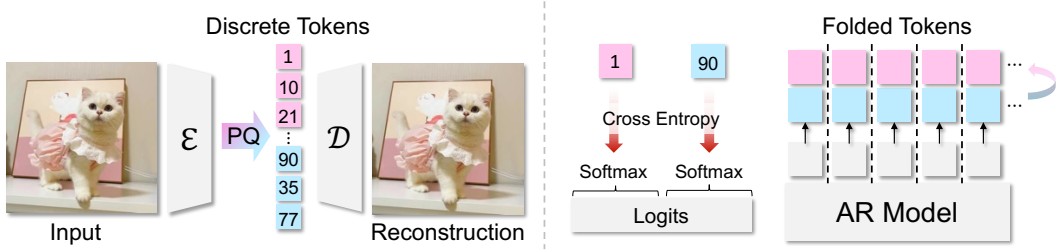

(a) Image tokenization with Product Quantization (PQ)     (b) Autoregressive (AR) modeling with folded tokens

Figure 1: Illustration of ImageFolder tokenizer and its corresponding autoregressive (AR) modeling with parallel prediction. (a) ImageFolder utilizes product quantization to obtain two sets of spatially aligned tokens that capture distinct aspects of images. (b) With the tokens from ImageFolder, AR models can predict two tokens from one logit thus significantly shortening the sequence length and benefiting the performance.

## Abstract

Image tokenizers are crucial for visual generative models, e.g., diffusion models (DMs) and autoregressive (AR) models, as they construct the latent representation for modeling. Increasing token length is a common approach to improve the image reconstruction quality. However, tokenizers with longer token lengths are not guaranteed to achieve better generation quality. There exists a trade-off between reconstruction and generation quality regarding token length. In this paper, we investigate the impact of token length on both image reconstruction and generation and provide a flexible solution to the tradeoff. We propose **ImageFolder**, a semantic tokenizer that provides spatially aligned image tokens that can be folded during autoregressive modeling to improve both generation efficiency and quality. To enhance the representative capability without increasing token length, we leverage dual-branch product quantization to capture different contexts of images. Specifically, semantic regularization is introduced in one branch to encourage compacted semantic information while another branch is designed to capture the remaining pixel-level details. Extensive experiments demonstrate the superior quality of image generation and shorter token length with ImageFolder tokenizer.

## 1 Introduction

Image generation (Chang et al., 2022; Dhariwal & Nichol, 2021; He et al., 2024; Esser et al., 2021; Yu et al., 2024d; Weber et al., 2024) has achieved notable progress empowered by diffusion models (DMs) (Dhariwal & Nichol, 2021; Rombach et al., 2022b; Peebles & Xie, 2023b) and autoregressive (AR) models/large language models (LLMs) (Vaswani et al., 2023). Different from diffusion models that leverage a continuous image representation (Li et al., 2024c; Dhariwal & Nichol, 2021; Rombach et al., 2022b; Peebles & Xie, 2023b), AR models and LLMs typically discretize image

---

[*]This work was done when Xiang Li was an intern at Adobe Research.

features into tokens (Yu et al., 2024b; Li et al., 2024d; Tian et al., 2024) and conduct the autoregressive modeling in the discrete space. Therefore, the performance of AR models/LLMs is largely influenced by the properties of the image tokenizers. This motivates the community to push the edge of image tokenizers for high-quality and efficient visual generation with large language models.

VQGAN (Esser et al., 2021) is a pioneer work on image tokenization that leverages a vector quantization (Gray, 1984) operation to map the continuous image feature to discrete tokens with a learnable codebook. Recently, several improvements from different perspectives have been made to the original VQGAN (Lee et al., 2022a; Yu et al., 2023b; Mentzer et al., 2023; Zhu et al., 2024a; Takida et al., 2023; Huang et al., 2023; Zheng et al., 2022; Yu et al., 2023a; Weber et al., 2024; Yu et al., 2024a; Luo et al., 2024). For example, VAR (Tian et al., 2024) proposes a multi-scale vector quantization (MSVQ) which quantizes images into a series of multi-scale tokens and conducts the autoregressive modeling in a scale-based manner, leading to a remarkable latency reduction and reconstruction improvement. However, in a trade-off of the speed and reconstruction quality, the sequence length of the quantized tokens is much longer compared to other models which could result in a larger training cost and difficulty of incorporating in LLMs with long contexts (Jin et al., 2024; Ding et al., 2024). SEED (Ge et al., 2023) and TiTok (Yu et al., 2024d) improve the VQGAN by injecting semantics into the quantized codewords. Since the semantic tokens can convey more compacted information about the image, they can achieve a much smaller token number to represent an image compared to the VQGAN. Nevertheless, a smaller token number typically leads to information loss. SEED (Ge et al., 2023) tokenizer discards all pixel-level details and only retains semantics to achieve a smaller token number. TiTok (Yu et al., 2024d) conducts concatenated tokenization with multiple quantization operations to achieve a higher compression rate while still suffering from detail distortion during reconstruction. Based on previous arts, we can summarize a **trade-off between reconstruction and generation**: (1) a longer token number can lead to a better reconstruction performance while making the sequence length too long for autoregressive generation (e.g., requiring larger model capacity (Sun et al., 2024), suffering more error propagation during AR sampling (Kaiser et al., 2018) and more training costs (Tian et al., 2024)), and (2) a smaller token number can lead to inferior reconstruction performance while benefit generation. Motivated by this, we aim to explore whether it is possible to retain a sufficient number of tokens for high-quality reconstruction without lengthening the sequence length for generation.

While it is possible to predict multiple tokens parallelly from one logit (Alexey, 2020; Peebles & Xie, 2023a) as shown in Fig. 1 (b). This predicting scheme ignores the dependency between tokens predicted in parallel in the AR modeling. However, as the vector quantization (Esser et al., 2021) is typically conducted on the spatial-preserved image features, each quantized token represents a spatial region of the original image, leading to high spatial dependency among tokens as shown in Fig. 2. To address this problem, we aim to maintain spatial dependency in AR modeling while constructing distinct tokens within each spatial location to obtain token pairs with low dependency using product quantization (Jegou et al., 2010).

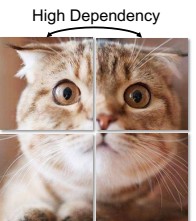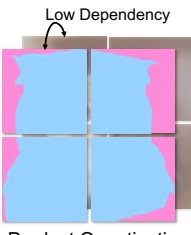

Figure 2: Token dependency.

In this paper, we propose **ImageFolder**, a semantic image tokenizer that provides spatially aligned image tokens, which can be folded for a shorter token length during autoregressive modeling. We demonstrate that even if the overall token number remains the same, the folded tokens demonstrate a better generation performance compared to unfolded ones. Specifically, different from previous tokenizers (Lee et al., 2022a; Yu et al., 2023b; Mentzer et al., 2023; Zhu et al., 2024a; Takida et al., 2023; Huang et al., 2023; Zheng et al., 2022), we leverage product quantization to separately capture different information of images, i.e., semantics and pixel-level details, with two branches. Within each branch, we use the multi-scale residual quantization as our quantizer. A quantizer dropout is further proposed to force each residual layer to represent an image with different bitrates, thus compensating the unmodeled dependency in the next-scale AR modeling (Tian et al., 2024). With the above designs, the spatially aligned tokens from ImageFolder can be folded for a parallel prediction during AR modeling as shown in Fig. 1 (b), which leads to half of the modeling token length while achieving a better generation quality. Our contribution can be summarized in threefold:

- We present ImageFolder, a novel image tokenizer that enables shorter token length during autoregressive modeling while not losing reconstruction/generation quality.

- We explore the product quantization in the image tokenizer. Moreover, we introduce semantic regularization and quantizer dropout to enhance the representation capability of the quantized image tokens.

- We investigate the trade-off between the generation and reconstruction regarding token length with extensive experiments, facilitating the understanding of image generation with AR modeling.

## 2 RELATED WORKS

**Image tokenizers.** Image tokenizer has seen significant progress in multiple image-related tasks. Traditionally, autoencoders (Hinton & Salakhutdinov, 2006; Vincent et al., 2008) have been used to compress images into latent spaces for downstream work such as (1) generation and (2) understanding. In the case of generation, VAEs (Van Den Oord et al., 2017; Razavi et al., 2019a) learn to map images to probabilistic distributions; VQGAN (Esser et al., 2021; Razavi et al., 2019b) and its subsequent variants (Lee et al., 2022a; Yu et al., 2023b; Mentzer et al., 2023; Zhu et al., 2024a; Takida et al., 2023; Huang et al., 2023; Zheng et al., 2022; Yu et al., 2023a; Weber et al., 2024; Yu et al., 2024a; Luo et al., 2024; Zhu et al., 2024b) introduce a discrete latent space for better compression for generation. On the other hand, understanding tasks, such as CLIP (Radford et al., 2021), DINO (Oquab et al., 2023; Darcet et al., 2023; Zhu et al., 2024c), rely heavily on LLM (Vaswani et al., 2023; Dosovitskiy et al., 2021) to tokenize images into semantic representations (Dong et al., 2023; Ning et al.). These representations are effective for tasks like classification (Dosovitskiy et al., 2021), object detection (Zhu et al., 2010), segmentation (Wang et al., 2021), and multi-modal application (Yang et al., 2024). For a long time, image tokenizers have been divided between methods tailored for generation and those optimized for understanding. After the appearance of (Yu et al., 2024c), which proved the feasibility of using LLM as a tokenizer for generation, some works (Wu et al., 2024) are dedicated to unify the tokenizer for generation and understanding due to the finding in (Gu et al., 2023).

**Diffusion models for image generation.** Diffusion models, initially introduced by Sohl-Dickstein et al. (Sohl-Dickstein et al., 2015) as a generative process and later expanded into image generation by progressively infusing fixed Gaussian noise into an image as a forward process. A model, such as U-Net (Ronneberger et al., 2015), is then employed to learn the reverse process, gradually denoising the noisy image to recover the original data distribution. In recent years, this method has witnessed significant advancements driven by various research efforts. Nichol et al. (Nichol & Dhariwal, 2021), Dhariwal et al. (Dhariwal & Nichol, 2021), and Song et al. (Song et al., 2022) proposed various techniques to enhance the effectiveness and efficiency of diffusion models, paving the way for improved image generation capabilities. Notably, the paradigm shift towards modeling the diffusion process in the latent space of a pre-trained image encoder as a strong prior (Van Den Oord et al., 2017; Esser et al., 2021) rather than in raw pixel spaces (Vahdat et al., 2021; Rombach et al., 2022b) has proven to be a more efficient and instrumental method for high-quality image generation. Moreover, a lot of research on the model architecture replaces or integrates the vanilla U-Net with a transformer (Peebles & Xie, 2023b) to further improve the capacity and efficiency of multi-model synthesis on diffusion model. Inspired by these promising advancements in diffusion models, numerous foundational models have emerged, driving innovation in both image quality and flexibility. For instance, Glide (Nichol et al., 2021) introduced a diffusion model for text-guided image generation, combining diffusion techniques with text encoders to control the generated content. Cogview (Ding et al., 2021) leveraged transformer architectures alongside diffusion methods to enhance image generation tasks. Make-A-Scene (Gafni et al., 2022) and Imagen (Saharia et al., 2022) focused on high-fidelity image synthesis conditioned on textual inputs, showcasing the versatility of diffusion models across modalities. DALL-E (Ramesh et al., 2021) and Stable Diffusion (Rombach et al., 2022a) brought diffusion models to mainstream applications, demonstrating their ability to generate high-resolution photorealistic images. Additionally, recent models such as MidJourney (MidJourney Inc., 2022) and SORA (OpenAI, 2024) have further refined the use of diffusion models in creative and commercial contexts, illustrating the growing influence and adaptability of these models in a range of domains. Even though the performance of image generation in diffusion models

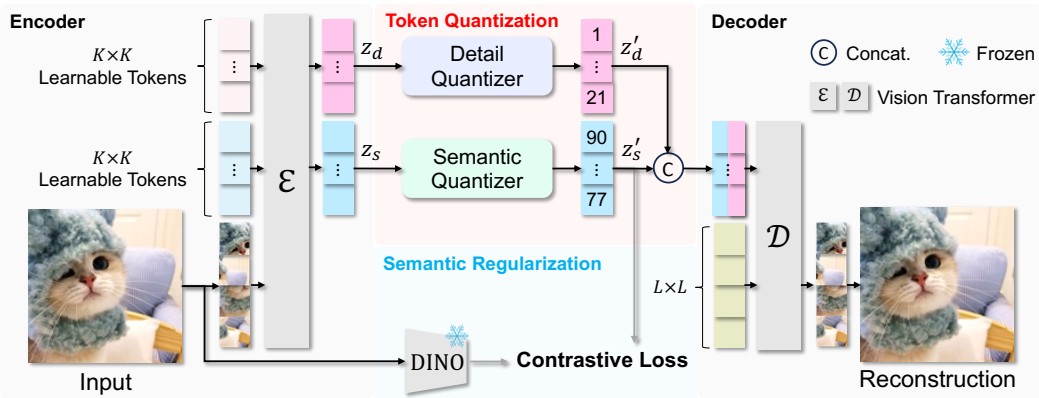

Figure 3: Overview of ImageFolder. ImageFolder leverages vision transformers (Alexey, 2020) to encode and decode images. Given an image, two sets of $K \times K$ learnable tokens are used to generate spatially-aligned low-resolution features from the image. After that, a product quantization is used to obtain discrete image representation. A semantic regularization is applied in one of the quantizers to inject semantic constraints. The quantized tokens are concatenated to serve as input for the image decoder to reconstruct images.

has shown impressive results, the training cost and inference speed remain bottlenecks, motivating the exploration of more efficient approaches such as leveraging large language models (LLMs) for image generation.

**Autoregressive visual generation.** Autoregressive models have shown remarkable success in generating high-quality images by modeling the distribution of pixels or latent codes in a sequential manner. Early autoregressive models such as PixelCNN (Van den Oord et al., 2016) pioneered the approach of predicting pixel values conditioned on previously generated pixels. The transformer architecture (Vaswani et al., 2023), first proposed in NLP, has spread rapidly to image generation (Shi et al., 2022; Mizrahi et al., 2024) because of its scalability and efficiency. MaskGIT (Chang et al., 2022) accelerated the generation by predicting tokens in parallel, while MAGE (Li et al., 2023) applied MLLM (Bao et al., 2022; Peng et al., 2022) to unify the visual understanding and the generation task. Recently, autoregressive models continued to show their scalability power in larger datasets and multimodal tasks (He et al., 2024); models like LlamaGen (Sun et al., 2024) adapting Llama (Touvron et al., 2023) architectures for image generation. New directions such as VAR (Tian et al., 2024; Li et al., 2024d), MAR (Li et al., 2024c) and Mamba (Li et al., 2024a) have further improved flexibility and efficiency in image synthesis. Currently, more and more unified multimodal models like SHOW-O (Xie et al., 2024), Transfusion (Zhou et al., 2024) and Lumina-mGPT (Liu et al., 2024) continue to push the boundaries of autoregressive image generation, demonstrating scalability and efficiency in diverse visual tasks.

## 3 METHOD

**Preliminaries.** Product quantization (PQ) (Jegou et al., 2010; Guo et al., 2024) aims to quantize a high-dimension vector to a combination of several low-dimension tokens, which has shown a promising capability to capture different contexts across sub-quantizers (Baevski et al., 2020). Given a continuous feature $z$, product quantization performs as:

$$\mathcal{P}(z) = \mathcal{Q}_1(z_1) \oplus \cdots \oplus \mathcal{Q}_P(z_P), \mathcal{Q}_p \sim \mathcal{C}_p, \tag{1}$$

where $\oplus$ denotes channel-wise concatenation (cartesian product), $\sim$ denotes that vector quantizer $\mathcal{Q}_p$ is associated with the codebook $\mathcal{C}_p = \{e_j\}_{j=1}^J$, i.e., $\mathcal{Q}_p$ maps a feature $z_p$ to a codeword $e_p = \arg\min_{e_j \in \mathcal{C}_p} \|z_p - e_j\|_2^2$ that minimizes the distance between $z_p$ and $e_j \in \mathcal{C}_p$. $P$ is the product number, i.e., sub-vector number and $z_p$ is a sub-vector from $z$ having $z = \oplus_{p=1}^P z_p$. Product quantization first divides the target vector into several sub-vectors and then quantizes them separately. After the quantization, the quantized vectors resemble the original vector by concatenation.

## 3.1 IMAGEFOLDER

Imagefolder is a semantic image tokenizer that tokenizes an image into spatially aligned semantic and detail tokens with product quantization. The Imagefolder is designed based on two preliminary observations. (1) As preliminary experiments shown in Tab 1, increasing the token number generally leads

| # Tokens | $10 \times 10$ | $12 \times 12$ | $14 \times 14$ |
|---|---|---|---|
| rFID | 4.15 | 3.21 | 2.40 |
| gFID | 5.22 | 5.34 | 6.24 |

Table 1: Performance against #Tokens.

to better reconstruction quality. However, generation behaves differently — longer token sequences can result in inferior outcomes.

(2) Semantic tokenizers, e.g., SEED (Ge et al., 2023) and Titok (Yu et al., 2024d) can encode an image into only a few tokens (32 tokens). As shown in Fig. 4, even though losing pixel-level structure/details of the image, the reconstructed images are semantically correct spatially. This motivates us to leverage semantics to obtain compacted image representation while encoding the remaining image details with additional tokens.

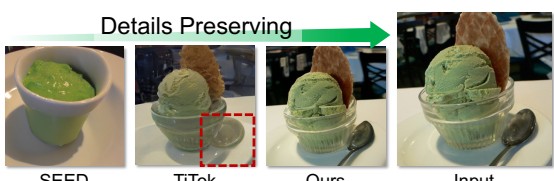

Figure 4: Image reconstruction.

**Architecture.** As shown in Fig. 3, given an input image $I$, we first patchify it into a set of $L \times L$ tokens where $L$ is the patch size. After that, the image tokens are concatenated with two sets of $K \times K$ learnable tokens and serve as the input to the transformer encoder $\mathcal{E}$. The same spatial positional encodings are added to the learnable tokens to inform the spatial alignment. A level embedding is additionally used to convey the difference across $K \times K$ tokens. Let us denote the encoded tokens corresponding to the learnable tokens as $z_d, z_s$. To discretize them, we use a detail quantizer $\mathcal{Q}_d$ and a semantic quantizer $\mathcal{Q}_s$ to conduct product quantization. The quantized tokens $z'_d$ and $z'_s$ are further concatenated with a set of $L \times L$ learnable tokens to decode the reconstructed image with a decoder $\mathcal{D}$.

**Token quantization.** As shown in Fig. 3, we leverage product quantization to quantize an image with semantics and pixel-level details using two branches separately. Specifically, semantic regularization is applied to the semantic branch to guide the semantics and distinguish it from the detail branch. Moreover, a quantizer dropout strategy is used to facilitate the quantizer to learn multi-scale image representation.

(1) **Quantizer dropout**: We use multi-scale residual quantizer (MSRQ) (Tian et al., 2024) for each branch, leading to multi-resolution tokens $z'_1, z'_2, \cdots, z'_N$ where $N$ denotes the number of residual steps. In the $i$-th residual step, the MSRQ quantizer first downsamples the input to a smaller $K_i \times K_i$ resolution and then quantizes it by mapping it to its closest vector in the codebook $\mathcal{C} = \{e_j\}_{j=1}^J$ as $\hat{z}_i[a, b] =$

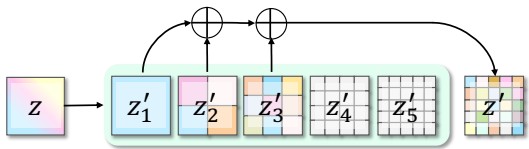

Figure 5: Illustration of quantizer dropout in multi-scale residual quantizer.

$\arg\min_{e_j \in \mathcal{C}} \|z_i[a, b] - e_j\|_2^2$, where $[a, b]$ is the spatial coordinates. After the nearest-neighbor look-up, the quantized representation $\hat{z}_i$ is up-sampled back to the original resolution $K \times K$ with a convolution to handle the blurring as:

$$z'_i = \gamma \times \text{conv}(\hat{z}_i) + (1 - \gamma) \times \hat{z}_i, \tag{2}$$

where $\gamma$ is set to a constant scalar as $0.5$. Specifically, to enhance the representation capability, we notice that a quantizer dropout strategy (Kumar et al., 2024; Zeghidour et al., 2021) can largely improve generation performance. During training, as shown in Fig. 5, we randomly drop out the last several quantizers. The quantizer dropout happens with a ratio of $p$. With the quantizer dropout, the final output of the MSVQ can be formulated as:

$$z' = \sum_{i=1}^{n} z'_i, N_{start} \leq n \leq N. \tag{3}$$

Applying quantizer dropout enables residual quantizers to encode images into different bitrates depending on the residual steps. This will significantly smooth the next-scale autoregressive prediction, as we will show in Fig. 7. To ensure training stability, the first $N_{start} = 3$ quantizers with extreme low resolution are never dropped out.

(2) **Product quantization**: We quantize images with separate MSVQ to capture different information. Denoting the output of detail and semantic quantizer as $z'_d$ and $z'_s$ respectively, we concatenate them in a channel-wise manner $z'_d \oplus z'_s$ to form the final quantized token. It is worth noting that both quantizers share the same quantizer dropout setting for each sample. The codebooks are updated separately for each quantizer.

**Semantic regularization.** To impose semantics into the tokenized image representation, we propose a semantic regularization term to the quantized token $z'_s$. A frozen pre-trained DINOv2 model (Oquab et al., 2023) is utilized to extract the semantic-rich visual feature $f_s$ of the input image $I$. We first pool the quantized token $z'_s$ to $1 \times 1$. After that, a CLIP-style contrastive loss (Radford et al., 2021) is used to perform visual alignment: maximizing the similarity be-

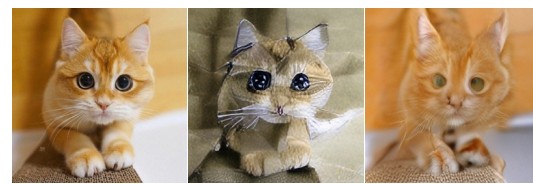

Input          w./o. semantic          w./o. detail

Figure 6: Visualization of zero out $z'_s$ or $z'_d$.

tween the quantized tokens $z'_s$ and their corresponding DINO representation $f_s$, while minimizing the similarity between $z'_s$ and other representations $f_s$ from different images within one batch. To facilitate semantic learning, we initialize the image encoder $\mathcal{E}$ with the same DINOv2 weight as the frozen one. We notice that the tokens with quantizer dropout will lead to instability during training. In this way, the contrastive loss is only applied to the tokens without being applied dropout. As shown in Fig. 6, we visualize reconstructed images with zero $z'_s$ or $z'_d$ to investigate the information captured by different quantizers. When we zero out the semantic tokens $z'_s$, we notice that the boundary of the reconstructed image remains aligned with the input. In contrast, zeroing out detail tokens reconstructs an image that is more similar to the input while the object identity is not maintained (like what SEED (Ge et al., 2023) and TiTok (Yu et al., 2024d) perform).

We provide a more in-depth discussion here to facilitate the understanding of adopting semantic regularization only in one branch. We notice a concurrent work VILA-U (Wu et al., 2024) also aims to inject

| Tuning | Linear | LORA | Full |
|--------|--------|------|------|
| rFID   | >5     | 3.55 | 1.57 |

Table 2: rFID of DINOv2 as tokenizer.

semantics into the quantized tokens while their design results in performance degradation for image generation. As shown in Tab 2, we compare the rFID with pre-trained DINOv2 (Oquab et al., 2023) as an image encoder. We can find that the semantic-rich features from DINOv2 are not suitable for image reconstruction. We consider this can be because the features only contain structural semantics while discards pixel-level details. In this way, directly injecting semantics into all quantized tokens will force the image tokenizer to ignore high-frequency details. To address this issue, we only use semantic regularization in one branch and leave another branch to capture other information required to reconstruct the image.

**Loss function.** The ImageFolder is trained with composite losses including reconstruction loss $\mathcal{L}_{recon}$, vector quantization loss $\mathcal{L}_{VQ}$, adverserial loss $\mathcal{L}_{ad}$, Perceptual loss $\mathcal{L}_P$, and CLIP loss $\mathcal{L}_{clip}$:

$$\mathcal{L} = \lambda_{recon}\mathcal{L}_{recon} + \lambda_{VQ}\mathcal{L}_{VQ} + \lambda_{ad}\mathcal{L}_{ad} + \lambda_P\mathcal{L}_P + \lambda_{clip}\mathcal{L}_{clip}. \tag{4}$$

Specifically, the reconstruction loss measures the $L_2$ distance between the reconstructed image and the ground truth; vector quantization loss encourages the encoded features and its aligned codebook vectors; adversarial loss, applied from a PatchGAN (Isola et al., 2018) discriminator trained simultaneously, ensures that the generated images are indistinguishable from real ones; perceptual loss compares high-level feature representations from a pre-trained LPIPS (Zhang et al., 2018) to capture structural differences; and CLIP loss performs semantic regularization between semantic tokens and the pre-trained DINOv2 (Oquab et al., 2023) features. A LeCam regularization (Tseng et al., 2021) is applied to the adversarial loss to stabilize the training.

| ID | Method | Length | Evaluation Metric | |
|---|---|---|---|---|
| | | | rFID↓ | gFID↓ |
| 1 | Multi-scale residual quantization (Tian et al., 2024) | 680 | 1.92 | 7.52 |
| 2 | + Quantizer dropout | 680 | 1.71 | 6.03 |
| 3 | + Smaller quantized token size $K = 11$ | 286 | 3.24 | 6.56 |
| 4 | + Product quantization & Parallel decoding | 286 | 2.06 | 5.96 |
| 5 | + Semantic regularization on all branches | 286 | 1.97 | 5.21 |
| 6 | + Semantic regularization on one branch | 286 | 1.57 | 3.53 |
| 7 | + DINO discriminator | 286 | 0.80 | 2.60 |

Table 3: Ablation study on the improvement of ImageFolder. We evaluate the rFID and gFID on the ImageNet validation set. DINO discriminator denotes the discriminator used in VAR's (Tian et al., 2024) tokenizer. ↑ and ↓ denote the larger the better and the smaller the better, respectively.

## 3.2 AUTOREGRESSIVE MODELING

To demonstrate the effectiveness of the proposed ImageFolder tokenizer, we train an autoregressive (AR) model to perform image generation based on the tokens obtained from ImageFolder. We follow VAR (Tian et al., 2024) to conduct the next-scale AR modeling for a faster inference speed. Recall that, we leverage product quantization to achieve spatially aligned tokens. Different from regular AR modeling which predicts one token from one logit, as shown in Fig. 1, we predict two tokens from the same logit with separate softmax operations. During training, the two predicted tokens are separately supervised with the corresponding ground truth. During inference, the tokens are sampled in parallel with a top-k-top-p sampling and then fed to the image decoder for image reconstruction. With our parallel predicting scheme, the visual tokens can be folded as input to the AR model, leading to half reduction of the original token length. We show that a shorter token length can benefit the generation quality, even though the overall token number remains the same.

**Parallel decoding.** We discuss the nature and differences between parallel decoding and traditional AR modeling. In vanilla AR models, given two tokens $x$ and $y$ (Fig. 1 (b)), the joint distribution $p(x, y)$ is modeled through the conditional dependency $p(x \mid y)p(y)$, meaning $x$ is sampled based on $y$. In contrast, our approach samples the joint distribution in parallel, assuming

$$p(x, y) = p(x)p(y), \tag{5}$$

which imposes independence between tokens $x$ and $y$. This independent assumption makes most existing image tokenizers, such as (Esser et al., 2021; Tian et al., 2024), unsuitable for parallel decoding as the spatial dependency is strong. To address this, we introduce *product quantization*, a technique commonly used to quantize high-dimensional spaces into several low-dimensional subspaces to reduce the dependencies. Specifically, we apply semantic regularization to one quantization branch while allowing the other branch to learn complementary information. This encourages that tokens from different branches capture distinct aspects of the image, resulting in weaker dependencies compared to the traditional spatial dependency among tokens.

In addition, with the quantizer dropout, each residual step can represent images with different bitrates. During the next-scale AR modeling, although the dependency within the current scale is not considered, the conditional dependency can still be refered from the low-resolution tokens in previous AR steps.

## 4 EXPERIMENTS

### 4.1 EXPERIMENTAL SETTINGS

We test our ImageFolder tokenizer on the ImageNet 256x256 reconstruction and generation tasks.

**Metrics.** We utilize Fréchet Inception Distance (FID) (Heusel et al., 2017), Inception Score (IS) (Salimans et al., 2016), Precision, and Recall as metrics for assessing the image generation.

**Implementation details.** For the ImageFolder tokenizer, if there is no other specification, we follow the VQGAN training recipe of LlamaGen (Sun et al., 2024). We initialize the image encoder

| Type | Method | Tokenizer | | Generator | | | | | | |
|------|--------|-----------|------|-----------|------|------|------|------|------|------|
| | | rFID↓ | L.P.↑ | gFID↓ | IS↑ | Pre↑ | Rec↑ | #Para | Leng. | Step |
| Diff. | ADM (Dhariwal & Nichol, 2021) | - | - | 10.94 | 101.0 | 0.69 | 0.63 | 554M | - | 1000 |
| Diff. | LDM-4 (Rombach et al., 2022b) | - | - | 3.60 | 247.7 | - | - | 400M | - | 250 |
| Diff. | DiT-L/2 (Peebles & Xie, 2023b) | 0.90 | - | 5.02 | 167.2 | 0.75 | 0.57 | 458M | - | 250 |
| Diff. | MAR-B (Li et al., 2024c) | 1.22 | - | 2.31 | 281.7 | 0.82 | 0.57 | 208M | - | 64 |
| NAR | MaskGIT (Chang et al., 2022) | 2.28 | - | 6.18 | 182.1 | 0.80 | 0.51 | 227M | 256 | 8 |
| NAR | RCG (cond.) (Li et al., 2024b) | - | - | 3.49 | 215.5 | - | - | 502M | 256 | 250 |
| NAR | TiTok-S-128 (Yu et al., 2024d) | 1.52 | 50.5 | 1.94 | - | - | - | 177M | 128 | 64 |
| NAR | MAGVIT-v2 (Yu et al., 2024b) | 0.90 | - | 1.78 | 319.4 | - | - | 307M | 256 | 64 |
| AR. | VQGAN (Esser et al., 2021) | 7.94 | - | 18.65 | 80.4 | 0.78 | 0.26 | 227M | 256 | 256 |
| AR. | RQ-Transformer (Lee et al., 2022b) | 1.83 | - | 15.72 | 86.8 | - | - | 480M | 1024 | 64 |
| AR. | LlamaGen-L (Sun et al., 2024) | 2.19 | 2.6 | 3.80 | 248.3 | 0.83 | 0.52 | 343M | 256 | 256 |
| AR. | VAR* (Tian et al., 2024) | 0.90 | 11.3 | 3.30 | 274.4 | 0.84 | 0.51 | 310M | 680 | 10 |
| AR. | ImageFolder (ours) | 0.80 | 58.0 | 2.60 | 295.0 | 0.75 | 0.63 | 362M | 286 | 10 |

Table 4: Performance comparison on class-conditional ImageNet 256x256. * denotes the utilized image tokenizer is trained on OpenImage (Kuznetsova et al., 2020). ↑ and ↓ denote the larger the better and the smaller the better, respectively. L.P. denotes linear probing accuracy on the ImageNet val set. The codebook utilization rate for the Imagefolder tokenizer is 100%.

with the weight of DINOv2-base. We use a cosine learning rate scheduler with a warmup for 1 epoch and a start learning rate of 3e-5. We set the quantizer drop ratio to 0.1. We set $\lambda_{clip} = 0.1$, $\lambda_{recon} = \lambda_{VQ} = \lambda_P = 1$ and $\lambda_{ad} = 0.5$. We set the residual quantizer scales to $[1, 1, 2, 3, 3, 4, 5, 6, 8, 11]$ (in total 286 tokens). The codebook size for each tokenizer is set to 4096. The residual quantization of each branch shares the same codebook across scales. For the generator, we adopt VAR's (Tian et al., 2024) GPT-2-based (Radford et al., 2019) architecture. We double the channel of the output head to predict two tokens in parallel.

## 4.2 PERFORMANCE ANALYSIS

**Roadmap to improve the ImageFolder tokenizer.** We first discuss the core designs of Image-Folder. (1) We begin with our baseline using multi-scale residual quantization from VAR (Tian et al., 2024) with a latent resolution of $16 \times 16$ and incorporate the proposed modules step by step. (2) As shown in Tab 3, with a quantizer dropout, gFID achieves a significant improvement of 1.49 FID. We consider this to be due to the smoother next-scale AR modeling as quantizer dropout will force each scale to represent the images with different resolutions. (3) After that, we adjust the latent resolution to $11 \times 11$ which leads to a performance drop for both reconstruction and generation due to a smaller token number. (4) To reduce the token length without decreasing the token number, we adopt a product quantization with corresponding parallel decoding. This modification results in a rFID of 2.06 and a gFID of 5.96 which outperforms the $16 \times 16$ counterpart's gFID of 6.03. (5) We further employ semantic regularization to enrich the semantics in the latent space for a more compact representation. (6) With the analysis of Tab 2, we realize that pure semantics is not enough for image reconstruction. Therefore, we adjust the semantic regularization to only one branch of the product quantization. Significant performance improvement is observed in both reconstruction and generation. Regularization on a single branch encourages different branches to capture different information about the image for a better reconstruction quality and reduces the feature dependency thus leading to a more robust sampling during generation. With all components, the VAR with ImageFolder achieves 3.53 gFID with only 286 token lengths.

**Image generation comparison.** As shown in Tab 4, we compare ImageFolder with state-of-the-art generative models. We notice ImageFolder outperforms LlamaGen which shares a similar tokenizer training scheme. Compared with VAR (with advanced tokenizer training recipe, e.g., OpenImage, longer training, and stronger discriminator), our method achieves a comparable performance. In addition, the shorter token length of ImageFolder (286 length) also remarkably saves training/inference costs compared to VAR (680 length) which is $\mathcal{O}(n^2)$ to the length $n$. Since VAR (Tian et al., 2024) did not release their tokenizer training code and recipe, we can not reproduce their result. With our architecture and training recipe, the vanilla MSVQ (Tian et al., 2024) only leads to a rFID of 1.92 and a gFID of 7.52.

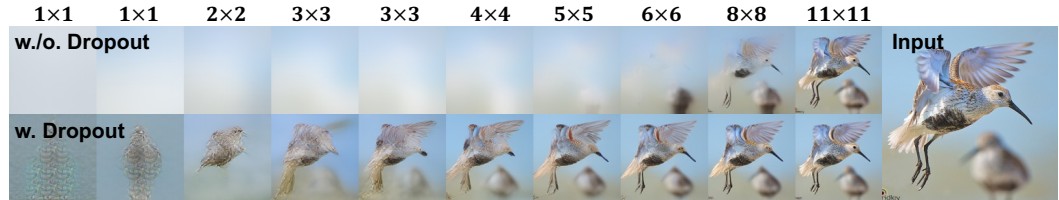

| 1×1 | 1×1 | 2×2 | 3×3 | 3×3 | 4×4 | 5×5 | 6×6 | 8×8 | 11×11 | |
|---|---|---|---|---|---|---|---|---|---|---|
| w./o. Dropout | | | | | | | | | | Input |
| w. Dropout | | | | | | | | | | |

Figure 7: Visualization of reconstructed images from different residual quantization steps. The residual scales used for this visualization is $[1, 1, 2, 3, 3, 4, 5, 6, 8, 11]$. The quantizer dropout starts from the third scale $2 \times 2$.

| $N_{start}$ | 1 | 3 | 5 |
|---|---|---|---|
| rFID | >20 | 1.57 | 1.69 |

(a) **Start dropout**.

| $p$ | 0 | 0.1 | 0.5 |
|---|---|---|---|
| rFID | 1.73 | 1.57 | 1.94 |

(b) **Dropout ratio**.

| $P$ | 1 | 2 | 4 |
|---|---|---|---|
| rFID | 2.74 | 1.57 | 0.97 |

(c) **Branch number**.

| $\lambda_{clip}$ | 0 | 0.01 | 0.1 |
|---|---|---|---|
| rFID | 1.97 | 1.67 | 1.57 |

(d) **Semantic Loss**.

Table 5: **Design choices for ImageFolder.**

### 4.3 MORE ANALYSIS

**Reconstruction of different residual steps.** We demonstrate the visualization of reconstructed images with different residual steps in Fig. 7. Without quantizer dropout, the core information is mainly concentrated on the last several layers. In contrast, with quantizer dropout, the one residual quantizer can perform image tokenization with different bitrates based on the residual step leading to progressively improved representation for the next-scale AR modeling.

**Conditional image generation.** Inspired by the teacher-forcing guidance proposed by ControlVAR (Li et al., 2024d), we teacher-force detail tokens of a reference image to conduct zero-shot conditional image generation with an AR model trained with ImageFolder. Specifically, we first encode the reference image to obtain the detail tokens. After that, we teacher-force the detail token during the AR modeling to conduct the conditional image generation (Li et al.,

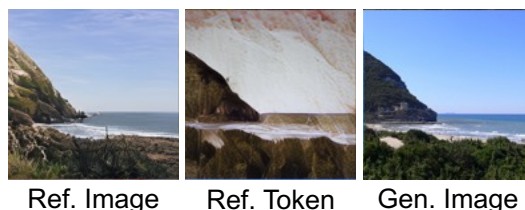

Ref. Image     Ref. Token     Gen. Image

Figure 8: Conditional image generation.

2024d). As shown in Fig. 8, we visualize the reference image, the reconstruction of the reference detail token and the generated image. We notice that the generated image follows the spatial shapes of the reference image while containing different visual contents, e.g., rocky mountain to frosty mountain. This experiment demonstrates a promising novel usage of ImageFolder on conditional image generation.

**Linear probing.** We further analyze the performance of the ImageFolder tokenizer by conducting linear probing on the ImageNet val set. As shown in Tab 4, we compare the linear probing top-1 accuracy of ImageFolder with LlamaGen and VAR. We notice that ImageFolder achieves a significantly superior linear probing accuracy compared to comparable methods. We notice that VAR also demonstrates a decent linear probing accuracy which can be credited to its advanced training with the DINO discriminator.

**Inference latency.** We compare the inference speed of LlamaGen, VAR, and our method, as summarized in Tab 6. LlamaGen exhibits slower inference due to its next-token autoregressive (AR) mechanism. Our approach achieves comparable latency to VAR, as both methods require the same number of AR steps (without CPU optimization). However, it is important to highlight that our method operates with approximately $\frac{1}{4}$ of the FLOPs compared to VAR, indicating a significant computational efficiency.

| Method | LlamaGen | VAR | Ours |
|---|---|---|---|
| Time (s) | 8.851 | 0.134 | 0.130 |

Table 6: Inference latency with $batchsize$=1 on single A100 GPU.

**Design choices.** We further demonstrate the impact of several design choices for our Imagefolder. As Tab 5 shown, (1) quantizer dropout can significantly improve the capacity of our tokenizer, but

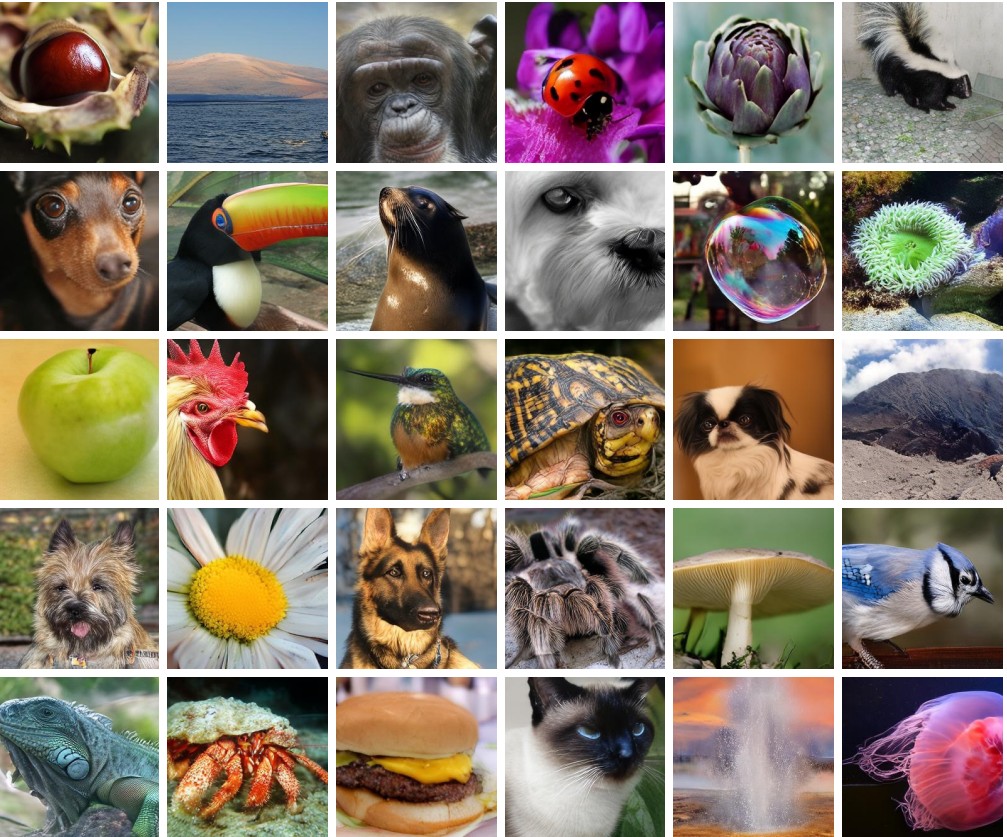

Figure 9: Visualization of $256 \times 256$ image generation task within ImageNet classes.

if we drop too many quantizers during training, it will unstabilize the training of our tokenizer and cause degradation. (2) Quantizer dropout will force the model to reconstruct the image with different bitrates while a too-large ratio will make the model focus too much on the intermediate outputs thus impacting the reconstruction of the last layer. (3) Increasing the number of branches can further improve the reconstruction quality whereas too many branches in the product quantizer also harm the quality of our generation, e.g., 5.24 gFID for 4 branch quantizer. (4) The use of semantic loss can facilitate the image quality for our tokenizer.

**Visualization.** We demonstrate qualitative visualization of the generative model trained with ImageFolder as shown in Fig. 9. The classes of generated images are among the ImageNet (Deng et al., 2009) dataset with a resolution of $256 \times 256$.

## 5   CONCLUSION

**Limitations.** Though the effectiveness of ImageFolder has been verified with extensive experiments, the tokenizer performance can be further improved with a more advanced training scheme which will be our focus in the short future.

In this paper, we introduced **ImageFolder**, a novel semantic image tokenizer designed to balance the trade-off between token length and reconstruction quality in autoregressive modeling. By folding spatially aligned tokens, ImageFolder achieves a shorter token sequence while maintaining high generation and reconstruction performance. Through our innovative use of product quantization and the introduction of semantic regularization and quantizer dropout, we demonstrated significant improvements in the representation of image tokens. Our extensive experiments have validated the effectiveness of these approaches, shedding light on the relationship between token length and visual generation quality. ImageFolder offers a promising direction for more efficient integration of visual generation tasks to AR models/large language models.

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

APPENDIX

## A    WHY DO WE NEED TOP-K-TOP-P SAMPLING?

Top-k and top-p sampling have proven to be effective methods for controlling randomness and diversity in the output of autoregressive (AR) models. In transformer-based image generation, top-k sampling selects tokens from the top $k$ most likely candidates at each step, truncating the probability distribution to avoid low-probability tokens that could degrade image quality. Top-p sampling, on the other hand, dynamically adjusts the sampling pool by selecting tokens whose cumulative probability exceeds a predefined threshold, ensuring a balance between flexibility and diversity in the generation process.

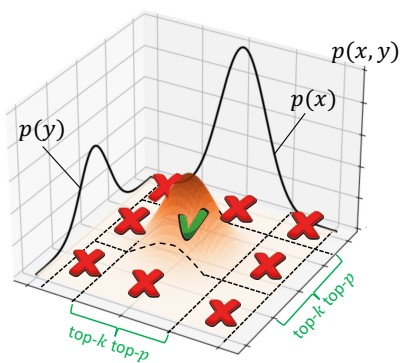

Figure 10: Top-k top-p sampling visualization

In our approach, we separately sample the semantic and detail tokens using top-k and top-p sampling from the same logit. Since this method assumes the independence of the two token types, we aim to prevent situations where one token is selected from a high-probability region while the other comes from a low-probability region. Specifically, as illustrated in Fig. 10, top-k and top-p sampling allows us to select tokens from regions where both semantic and detail tokens have high probabilities, ensuring consistent image quality.

## B    EVALUATION METRICS AND DATASET DETAILS

**Fréchet Inception Distance (FID)** (Heusel et al., 2017). FID measures the distance between real and generated images in the feature space of an ImageNet-1K pre-trained classifier (Szegedy et al., 2016), indicating the similarity and fidelity of the generated images to real images.

**Inception Score (IS)** (Salimans et al., 2016). IS also measures the fidelity and diversity of generated images. It consists of two parts: the first part measures whether each image belongs confidently to a single class of an ImageNet-1K pre-trained image classifier (Szegedy et al., 2016) and the second part measures how well the generated images capture diverse classes.

**Precision and Recall** (Kynkäänniemi et al., 2019). The real and generated images are first converted to non-parametric representations of the manifolds using k-nearest neighbors, on which the Precision and Recall can be computed. Precision is the probability that a randomly generated image from estimated generated data manifolds falls within the support of the manifolds of estimated real data distribution. Recall is the probability that a random real image falls within the support of generated data manifolds. Thus, precision measures the general quality and fidelity of the generated images, and recall measures the coverage and diversity of the generated images.

**ImageNet dataset.**    The ImageNet dataset (Deng et al., 2009) is a large-scale visual database designed for use in visual object recognition research. It contains over 14 million images, which are organized into thousands of categories based on the WordNet hierarchy. Each image is annotated with one or more object labels, making it a valuable resource for training and evaluating machine learning models in various computer vision tasks, such as image classification, object detection, and image generation. Its training set contains approximately 1.28 million images spanning 1,000 classes. Its validation set contains 50,000 images, with 50 images per class across the same 1,000 classes.

## C    VISUALIZATIONS

We provide additional visualizations as shown in Fig. 11 and Fig. 12.

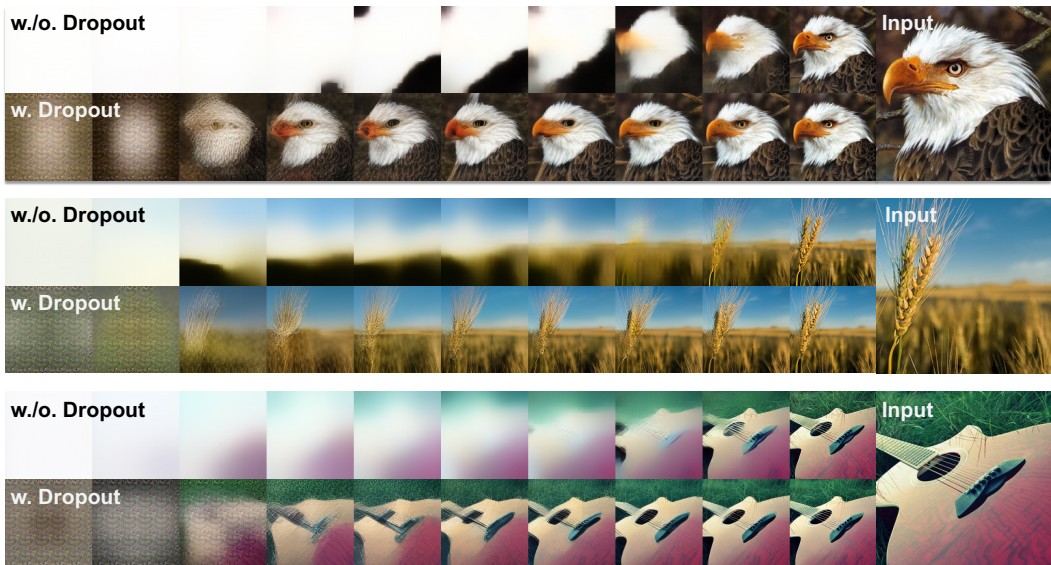

Figure 11: More visualization for demonstrating the effectiveness of quantizer dropout.

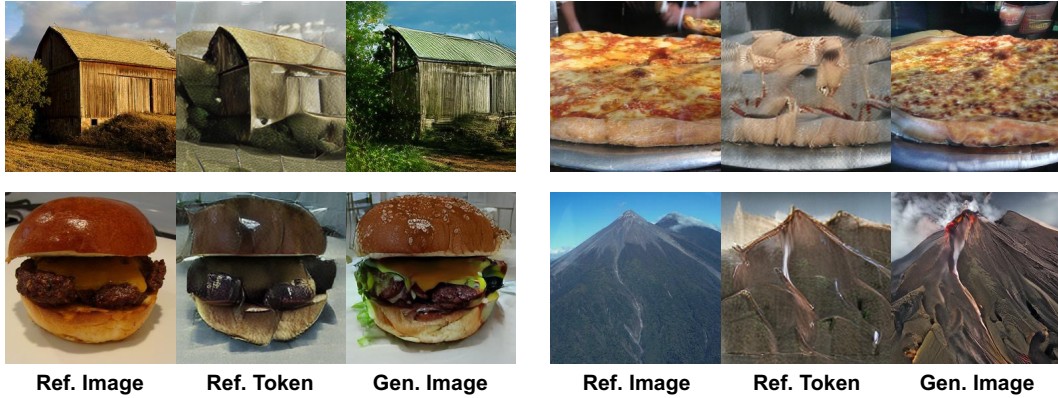

Figure 12: More visualization for zero-shot conditional image generation. During generation sampling, we teacher forcing all detail tokens to referenced detail tokens.

## D  TEACHER FORCING GUIDANCE FOR CONDITIONAL GENERATION

Classifier-free guidance has been proven to be effective in AR models (Chang et al., 2023) which take the same form as diffusion models as

$$p(I|C, c, c_t) \propto p(c|I, C, c_t)p(c_t|I, C)p(C|I)p(I).$$

In ControlVAR, we model the joint distribution of the controls and images. Therefore, we leverage a different extension of the probabilities as

$$p(c|I, C, c_t) = \frac{p(I, C|c, c_t)p(c, c_t)}{p(I, C|c_t)p(c_t)} = \frac{p(I, C|c, c_t)p(c)}{p(I, C|c_t)}$$

where $p(I, C|c, c_t)$ and $p(I, C|c, c_t)$ can be found from the output of ControlVAR. We follow previous works (Esser et al., 2021) to ignore the constant probabilities $p(c)$. By rewriting all terms with Baysian rule, we have

$$
\begin{aligned}
\log p(I|C, c, c_t) \propto\ & \log p(I) \\
& + \log p(I, C|c, c_t) - \log p(I, C|c_t) \\
& + \log p(I, C|c_t) - \log p(I, C) \\
& + \log p(I, C) - \log p(I).
\end{aligned}
$$

This corresponds to the image logits as

$$x^* = x(\uparrow\emptyset|\emptyset, \emptyset) + \gamma_{cls}(x(\uparrow C|c, c_t) - x(\uparrow C|\emptyset, c_t))$$
$$+ \gamma_{typ}(x(\uparrow C|\emptyset, c_t) - x(\uparrow C|\emptyset, \emptyset)) \tag{6}$$
$$+ \gamma_{pix}(x(\uparrow C|\emptyset, \emptyset) - x(\uparrow\emptyset|\emptyset, \emptyset))$$

where $\gamma_{cls}, \gamma_{typ}, \gamma_{pix}$ are guidance scales for controlling the generation.

## E   ADDITIONAL EXPERIMENTS

| ID | Branch1 | Branch2 | rFID | gFID |
|----|---------|---------|------|------|
| 1  | -       | -       | 2.06 | 5.96 |
| 2  | DINOv2  | DINOv2  | 1.97 | 5.96 |
| 3  | DINOv2  | -       | 1.57 | 3.53 |
| 4  | DINOv2  | SAM     | 1.47 | 4.05 |

Table 7: Ablation study on regularization.

**Regularization on more branches.**   We demonstrate the ablation study to investigate the performance of different encoders to conduct distillation. We notice that leveraging different encoders, i.e., DINO & SAM, to conduct regularization can lead to a superior rFID. However, we notice that the gFID does not improve as the rFID. Instead, it shows an inferior performance.

| Length | Scales | rFID | gFID |
|--------|--------|------|------|
| 265 | 1,1,2,3,3,4,5,6,8,10 | 1.86 | 3.98 |
| 286 | 1,1,2,3,3,4,5,6,8,11 | 1.57 | 3.53 |
| 309 | 1,1,2,3,3,4,5,6,8,12 | 1.22 | 3.51 |

Table 8: Ablation study on token length.

**Token length.**   We demonstrate an additional ablation study on token length where the patch size of the last scale increases from 10 to 12 leading to 265 to 309 token length for autoregressive modeling. We notice that the rFID keeps decreasing when the token length increases while the gFID is saturated for a token length near 286. Therefore, we utilize 286 token lengths in our final setting.

| Codebook size | 4096 | 8192 | 16384 |
|---------------|------|------|-------|
| rFID | 0.80 | 0.70 | 0.67 |

Table 9: Ablation study on codebook size.

**Token length.**   We conduct an ablation study on the codebook size of the ImageFolder tokenizer. We noticed that a large codebook size of 16384 makes a great improvement to the rFID to 0.67 which is comparable to the continuous tokenizers.

**Branch setting.**   We provide additional experiments complimentary to Table 3 to analyze the branch settings. We notice that in the setting of two branches with one for semantic information and one for detail information, the model achieves the best performance.

## F   PRODUCT QUANTIZED SPACE

Product quantization divides the original space into subspaces and then conducts quantization on the subspaces. As shown in Fig. 13, we demonstrate two scenarios of product quantized space with different data sample layouts.

As the product quantization leverages the cartesian product to connect the separately quantized subspaces, it will ensure a symmetric property of the quantized space (symmetric indicates the

| Method | Branch | rFID | gFID |
|---|---|---|---|
| Single detail branch | 1 | 3.24 | 6.56 |
| Single semantic branch | 1 | 2.79 | 5.81 |
| Two semantic branches | 2 | 1.97 | 5.21 |
| Two detail branches | 2 | 2.06 | 5.96 |
| One semantic, one detail | 2 | 1.57 | 3.53 |

Table 10: Comparison of different methods with the corresponding branch, rFID, and gFID values.

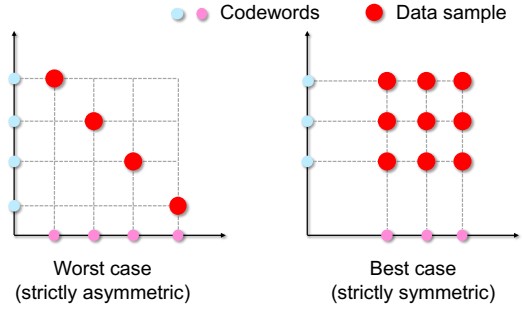

Figure 13: A visualization of a product quantized space with 2 subspaces.

quantized tokens are symmetric to the axes, showing a grid pattern). In this way, if the original data sample is strictly asymmetric, considering a two-subspace case (Fig. 13), it will require $2\times\#$sample codewords to perfectly encode all the samples. Differently, if the data samples are arranged in a symmetric manner, the required number of codewords can be largely reduced to $2 \times \sqrt{\#\text{sample}}$.

In this way, a potential improvement direction is to find subspaces that have desired properties to be quantized. In the ImageFolder framework, this can be achieved by leveraging contrastive loss to guide the latent space in different branches.

