# OpenReview forum: "ImageFolder: Autoregressive Image Generation with Folded Tokens"
_ICLR.cc/2025/Conference — ICLR 2025 Poster_

### Official Review · Reviewer_BcrL · 2024-11-02

**Soundness:** 3
**Presentation:** 3
**Contribution:** 2
**Rating:** 8
**Confidence:** 4

**Summary:**

The paper is well-structured, comprehensive, and easy to understand, making the proposed concepts accessible. It presents a novel approach to decoupling features using separated quantizers and manages both the separated quantizers and input features through product quantization. Overall, the paper offers valuable insights and design strategies for quantizer development, contributing to advancements in feature token quantization.

**Strengths:**

1. The paper is well-structured, comprehensive, and easy to understand, making the proposed concepts accessible.

2. It presents a novel approach to decoupling features using separated quantizers and manages both the separated quantizers and input features through product quantization.

3. This approach effectively mitigates the performance drop observed in previous work caused by incorporating semantic features.

4. The model achieves comparable performance to state-of-the-art (SOTA) methods, demonstrating the efficacy of the proposed method.

5. Overall, the paper offers valuable insights and design strategies for quantizer development, contributing to advancements in feature token quantization.

**Weaknesses:**

1. The results section lacks a comprehensive comparison with a broader range of current state-of-the-art VAE and autoregressive (AR) token quantizers. Including such comparisons would provide stronger validation of the method’s performance.

2. In the main paper, the VAR model training has not reached final results, only intermediate outcomes are presented. Showing the final results would strengthen the evaluation.

3. In terms of ablation studies, the paper would benefit from additional experiments that test the model with only the semantic branch or only the detail branch under the same experimental setting. Additionally, showing results where only a single branch incorporates semantic information would provide a more complete analysis of the method’s effectiveness.

4. Finally, presenting quantizer dropout—a popular regularization technique—as a primary contribution does not seem particularly solid.

**Questions:**

Q1: In Table 1, does “tokens” refer to the number of input tokens or codebook tokens?

Q2: The symbol “L” in “247 quantizer drop” can be confusing as it is also used for patch size in the previous context. Additionally, in Table 3, patch size is denoted as  K=1, which adds to the confusion. Could you clarify this section further? Also, please consider refining the symbol definitions.

Q3: In Figure 5, should the grid size double after each downsampling? Currently, it shows 1x1 to 2x2 to 3x3—how should this be understood? Why does the grid increase with the step size? Also, why does resolution increase from left to right? Is this sequence reversed? Normally, the input has the highest resolution, which decreases as the step deepens.

Q4: In line 273, is dropout necessary for a simple semantic quantizer? Is it possible to dropout only one branch during training, so that we may know which branch will be impacted much more?

Q5: Has contrastive loss been considered using CLIP’s prompt or image features to create a more robust text-image pre-trained model? The paper mentions that DINO’s semantic-rich feature isn’t suitable for image generation. Would using models like CLIP potentially alleviate this issue, making a separate semantic quantizer branch unnecessary?

Q6: In AR modeling, does dividing one token into two sub-tokens and computing losses for each mean we are increasing the token count without lengthening the token length? Line 408, mentions increasing the token count without enlarging the token length, yet line 340 states that a shorter token length benefits quality while keeping the token count the same. Could you clarify this part?

Q7: A smaller patch size ( k=1 ) impacts performance significantly. Is this necessary, and have you tried other designs or experiments around patch size?

Q8: Could you add a comparison of different tokenizer performances for better illustration, such as comparisons with different VAEs and token quantizers?

Q9: In the 4-branch quantizer, how is the contrastive loss applied? Does this only worsen gFID in your results?

---

> ### Author Response · Authors · 2024-11-17
> **Author response - Part 1**
>
> We thank the reviewer for the time and effort in reviewing our submission.
>
> ---
>
> **"W1-2. Lacks comparison to VAE and tokenizers. The VAR model training has not reached the final results."**
>
> We updated Table 4 in the revised submission to reflect the comparison with the previous tokenizers and demonstrate the final results of generation. Please note that the only change to the training recipe is that we adopt the stronger DINO discriminator used in the VAR tokenizer's training as shown in Table 3.
>
> - Notably, our tokenizer achieves a **0.8 rFID** which outperforms MAGVIT-V2 (0.9 rFID) as well as the tokenizer used in VAR (0.9 rFID).
> - Our tokenizer achieves a **58.0 linear probing accuracy** on the ImageNet val set, significantly outperforming 11.3 using VAR's tokenizer.
> - When using tokens from ImageFolder, the same GPT2 architecture used in VAR can achieve a **2.6 gFID** with a **10-step** inference schedule which significantly outperforms the 3.3 of the original VAR.
>
> ---
>
> **"W3. Additional experiments on only the semantic branch or only the detail branch under the same setting. A single branch incorporates semantic information."**
>
> We thank the reviewer for the suggestions. We consider the experiments about the suggested only semantic branch and only detail branch are available in Table 3 as ID 4 (only detail) and ID 5 (only semantic). We combine Table 3 and report the additional experiment of a single branch with semantic information as shown in Table A. We can notice that two branches with one for semantic and one for detail can lead to the best performance. We have incorporated the results in Table 8 of the revised version.
>
> | Method | Branch | rFID | gFID |
> |--------|--------|------|------|
> | Single detail  | 1   | 3.24 | 6.56 |
> | Single sem. | 1   |  2.79   | 5.81 |
> | All sem    | 2   | 1.97    | 5.21 |
> | All detail    | 2   | 2.06    | 5.96 |
> | One sem, one detail | 2 | 1.57 | 3.53 |
>
> Table A. Additional experiments about semantic settings.
>
> ---
>
> **"W4. Quantizer dropout"**
>
> We agree that quantizer dropout is not the core contribution of ImageFolder. Instead, we consider the major contribution of ImageFolder as (1) we explored the tradeoff between rFID and gFID and provided a solution to balance them, (2) we analyzed the impact of semantics to reconstruction and generation, and provided an approach to address both of semantics and details, and (3) extensive experiments and discussion towards a state-of-the-art image tokenizer (better rFID of 0.8 than MAGVIT-v2's 0.9 and VAR's 0.9) for generation.
>
> We explicitly mentioned quantizer dropout in the method section since the training of quantizer dropout requires several model-specific settings to make it functional. We will discuss the details in the response to Q4.
>
> ---
>
> **"Q1. "tokens" in Table 1"**
>
> We would like to clarify that "tokens" in Table 1 refer to input tokens.
>
> ---
>
> **"Q2. $L_i$ in Line 247. Patch size $K=11$ in Table 3."**
>
> Thanks for the suggestion. We would like to clarify that $L$ and $K$ denote the input patch size and quantized token size. We apologize that $L_i$ in Line 247 is a typo which should be $K_i$. To avoid potential misunderstanding, we use "quantized token size" to describe $K$ in Table 3 in the revision.
>
> ---
>
> **"Q3. Understanding of Fig 5 (MSVQ)."**
>
> In Fig 5, the grid size should keep increasing while it is not necessary to be doubled. As discussed in Line 377, we used [1,1,2,3,3,4,5,6,8,11] in ImageFolder as the final setting.
>
> We follow VAR to use multi-scale residual quantization (MSVQ) as our subquantizer. Specifically, MSVQ uses only 1 token to represent an image in the first residual layer and increases the token number with more residual depth. Combining quantizer dropout, different residual quantizer in MSVQ represents an image with different bitrates from low to high. This will lead to an image reconstruction from coarse to fine as shown in Fig 7. During the next-scale autoregressive prediction, the finer tokens are conditioned on the coarser tokens to gradually refine an image which permits a global understanding of the image compared to the traditional next-token prediction.
>
> ---
>
> **"Q4. Dropout."**
>
> Dropout is not necessary for a simple semantic quantizer but it can significantly improve the reconstruction and generation as demonstrated in Table 3. Empirically, we are not able to only dropout in one branch as it will lead to a severe stability issue and thus we made a note in Line 273. In addition, as denoted in Equ 3, the quantizer dropout should happen with a $N_{start}>3$ to ensure stability. We detailed those settings in the method section to enhance the reproducibility.

---

> ### Author Response · Authors · 2024-11-17
> **Author response - Part 2**
>
> **"Q5. CLIP v.s. DINO."**
>
> We would like to clarify that, as discussed in Line 301, we claimed that DINOv2's feature is not suitable for **reconstruction** as it loses pixel-level details. In our preliminary experiments, we compare DINO, CLIP (image encoder) and SAM as the distillation approach. As shown in Table B, DINO outperforms CLIP and SAM. We consider the necessity of an additional branch is to capture pixel-level details for reconstructing an image. Both the CLIP and DINO features are semantic-driven features that discard pixel-level details to boost understanding capability thus adopting CLIP features cannot alleviate the issue.
>
> | Method | Linear | LORA | Full |
> |--------|--------|------|------|
> | DINOv2   | $>$5   | 3.55 | 1.57 |
> | CLIP | $>$5   | -    | 1.65 |
> | SAM    | -      | -    | $>$5 |
>
>
> Table B. Extension of Table 2 with CLIP encoder.
>
> ---
>
> **"Q6. Tradeoff between token number and token length."**
>
> We apologize for the confusion. Both sentences (Line 408 and 340) aim to describe our core idea of reducing token length (in AR modeling) while keeping the token count. We have revised the expression in the revised version.
>
> ---
>
> **Q7. K=1 for the quantizer.**
>
> We would like to clarify that K=1 for the first scale is out of a generation consideration. The first scale with K=1 is not expected to have a satisfactory image quality. Instead, we hope it to be coarse and diverse enough to leave room for the subsequent steps to diversely enrich the details in a progressive manner as shown in Figure 7.
>
> We tried to start residual quantization from K=2 however this will lead to an inferior generation quality, i.e. 3.53 (K=1) v.s. 3.92 (K=2) gFID.
>
> ---
>
> **"Q8. Add tokenizer comparison."**
>
> Please refer to the response to Weakness 1-2.
>
> ---
>
> **"Q9. 4-branch quantizer."**
>
> In the results reported in Table 6(c), the contrastive loss is only applied in one branch while keeping the other three branches free.
>
> After submission DDL, we also tried a 3-branch setting, with two branches adding contrastive loss with two different pre-trained encoders and leaving one branch free. However, we still got >4 gFID. We did not observe a gFID gain with a branch number larger than 2.
>
> ---
>
> We hope our response has addressed your concerns. Please feel free to reach out with any additional questions.
>
> If you find our paper intriguing and promising, we would greatly appreciate it if you could consider raising your score to help increase the visibility of our work.

---

> > ### Comment · Reviewer_BcrL · 2024-11-25
> >
> > Thank you very much for the author’s response; my questions have been clearly answered. The paper is novel for proposing decoupling features using separated quantizers and manages both the separated quantizers and input features through product quantization. In the revised version, ImageFolder's results are comparable with SOTA, which de-risks its potential. Besides, several tricks are discussed in the paper to make the method possible to re-produce. If possible, I recommend the code to the public to inspire the community. I **raised my score to 8** for the clear response and novel method.

---

> > > ### Author Response · Authors · 2024-11-25
> > >
> > > Thank you for your kind words and support! We are in the final steps of preparing the code for release and look forward to sharing it soon.

---

### Official Review · Reviewer_NEtD · 2024-11-03

**Soundness:** 3
**Presentation:** 3
**Contribution:** 3
**Rating:** 6
**Confidence:** 4

**Summary:**

This paper introduces a novel image tokenizer called ImageFolder, which addresses the trade-off between image reconstruction quality and generation quality in visual generative models, such as diffusion models and autoregressive (AR) models. The key contributions of the paper can be summarized as follows: 1. The proposed tokenizer provides spatially aligned image tokens that can be folded during autoregressive modeling, resulting in a shorter token length without compromising reconstruction or generation quality. The experiments demonstrate that folded tokens outperform unfolded ones in terms of generation performance, even when the overall token count remains the same. 2. The paper explores the use of product quantization within the image tokenizer. It introduces semantic regularization and quantizer dropout to enhance the representational capability of the quantized image tokens by separately capturing semantic information and pixel-level details. 3. Extensive experiments are conducted to investigate the relationship between token length, generation quality, and reconstruction quality, providing deeper insights into the image generation process in AR modeling.

**Strengths:**

1.The ImageFolder tokenizer achieves better generation performance by effectively folding tokens, which enhances the efficiency of the autoregressive modeling process.

2.Despite reducing token length, the proposed method does not sacrifice the quality of image reconstruction, making it a balanced solution for generative tasks.

3.By leveraging product quantization and introducing mechanisms like semantic regularization and quantizer dropout, the tokenizer captures richer semantic and pixel-level information, leading to more accurate image representations.

4.The paper provides extensive experimental results that validate the effectiveness of the proposed method, offering a solid foundation for its claims and demonstrating its superiority over existing tokenizers.

5.The investigation into the trade-off between generation and reconstruction quality with respect to token length contributes to a better understanding of the dynamics involved in image generation, which can inform future research in the field.

**Weaknesses:**

1.The training is based on the tokenizer from LlamaGen. Is it possible to train one from scratch to better demonstrate its effectiveness?

2.How is the token length of 265 derived from the residual scales [1, 1, 2, 3, 3, 4, 5, 6, 8, 11]?

3.Are there other length design comparisons for token length, such as 265, 430, etc., to better validate the results?

**Questions:**

1.The training is based on the tokenizer from LlamaGen. Is it possible to train one from scratch to better demonstrate its effectiveness?

2.How is the token length of 265 derived from the residual scales [1, 1, 2, 3, 3, 4, 5, 6, 8, 11]?

3.Are there other length design comparisons for token length, such as 265, 430, etc., to better validate the results?

---

> ### Author Response · Authors · 2024-11-17
> **Author response**
>
> We thank the reviewer for the time and effort in reviewing our submission.
>
> ---
>
> **"W1/Q1. The training is based on LlamaGen. Scratch training?"**
>
> We would like to clarify that, as discussed in Line 373, our training hyperparameters/settings are based on the VQGAN training recipes of LlamaGen if no otherwise specification. However, the ImageFolder tokenizer is trained from scratch instead of loading the weights from LlamaGen.
>
> ---
>
> **"W2/Q2. Token length with scales [1, 1, 2, 3, 3, 4, 5, 6, 8, 11]."**
>
> We sincerely apologize for the typo. The token length for [1, 1, 2, 3, 3, 4, 5, 6, 8, 11] should be 286 = $1^2 + 2^2 + \cdots + 11^2$. The 265 is the token length for [1, 1, 2, 3, 3, 4, 5, 6, 8, 10]. We have updated the token length in the revised manuscript.
>
> ---
>
> **"W3/Q3. Ablation study on token length."**
>
> We report an ablation study on the token length. As shown in Table A, the token length of [1, 1, 2, 3, 3, 4, 5, 6, 8, 11] achieves the best tradeoff.
>
>
> | Length | Scales | rFID | gFID |
> |--------|--------|------|------|
> | 265    | 1,1,2,3,3,4,5,6,8,10  | 1.86 | 3.98 |
> | 286    | 1,1,2,3,3,4,5,6,8,10  | 1.57 | 3.53 |
> | 309    | 1,1,2,3,3,4,5,6,8,10  | 1.22 | 3.51 |
>
> Table A. Ablation study on the token length.
>
> We also did more experiments to investigate the scale setting for the first several tokens but we noticed that the scale settings used in VAR, i.e. [1,2,3,4,5,6,7,8,10,13,16] is a good practice. And our setting generally follows their token number for each scale. For example, VAR has 256=16x16 tokens and ImageFolder has 242=11x11x2 tokens in the last scale.
>
> ---
>
> We hope our response has satisfactorily addressed your concerns. Should you have any further questions, please do not hesitate to reach out.

---

### Official Review · Reviewer_w2WM · 2024-11-03

**Soundness:** 4
**Presentation:** 3
**Contribution:** 3
**Rating:** 6
**Confidence:** 4

**Summary:**

This paper proposes a novel method to solve the problem of how to improve image quality under finite length tokens, and proposes a semantic tokenizer, which realizes good image quality under smaller length tokens through Product Quantization.

**Strengths:**

1. The idea of this paper is novel. Through Product Quantization, token processing is carried out in smaller spaces, and good performance is achieved under fewer tokens.
2. The logic of this paper is rigorous, the argumentation is clear, the method is fully explained, and the experimental setting is rigorous.

**Weaknesses:**

Although this paper has reached an acceptable level, there are still several small problems that need to be worked on.
1. The description of performance indicators in the article is not in place. Tables 1, 2 and some later tables do not give a simple indication of superior performance represented by high or low indicators.
2. In Formula 4, the description of several hyperparameters and their Settings is missing.

**Questions:**

Please solve the problem I mentioned above

---

> ### Author Response · Authors · 2024-11-17
> **Author response**
>
> We thank the reviewer for the time and effort in reviewing our submission.
>
> ---
>
> **"W1. Performance indicators in the article are not in place, e.g. Table 1, 2"**
>
> We thank the reviewer for the suggestion. We have updated the description of performance indicators in the revised version.
>
> ---
>
> **"W2. Several hyperparameters and settings are missing for Equ 4."**
>
> We thank the reviewer for the suggestion. As discussed in Line 373, we followed the VQGAN training settings in LlamaGen for those hyperparameters. In the revised version, we explicitly provided a detailed description of hyperparameters for Equ 4.
>
> ---
>
> We hope our response can address your concerns. Your feedback has been constructive, and we sincerely thank you for your time and effort.

---

### Official Review · Reviewer_GK7c · 2024-11-04

**Soundness:** 3
**Presentation:** 3
**Contribution:** 2
**Rating:** 5
**Confidence:** 4

**Summary:**

- This paper presents a semantic image tokenizer that provides better token representations for visual autoregressive modeling, with improved generation quality and efficiency.

 - The authors leverage product quantization and semantic regularization for a more compact representation. Extensive experiments demonstrate the advantage of the proposed tokenizer over previous ones.

**Strengths:**

- This paper targets learning more semantically compact and disentangled image token representations, which leads to better autoregressive modeling with improved performance and shorter token length. The issue of information density(token length) and the explanation of previous image tokenizers have been hindering the performance of visual autoregressive modeling.

  - Compared to previous work on image tokenizers, this work investigates product quantization to separate different information about images. With the proposed muti-scale RQ, semantic regularization, and quantizer dropout, the representation of image tokens is enhanced.

 - Extensive experiments on autoregressive modeling and visualization demonstrate the superiority of ImageFold.

**Weaknesses:**

- Several important components in this work have been investigated in previous works, including adopting semantic regularization, multi-scale RQ, and parallel decoding has been explored in earlier works.

 - This work lacks either enough theoretical or empirical discussions about product quantization.  Additional details, such as more visualization and discussion as in Fig. 8, should be provided to validate the "disentangled" nature of the obtained image tokens.  (see Questions)

 - This work missed some important references. Although product quantization has been rarely discussed in image tokenizers. Previous works[1] on tokenizing other modalities such as audio have adopted relevant strategies.

[1] Addressing Index Collapse of Large-Codebook Speech Tokenizer with Dual-Decoding Product-Quantized Variational Auto-Encoder.

**Questions:**

- To better understand the method, the authors should add more analysis about the proposed tokenizer, such as the perplexity (codebook usage) with different techniques.

 -  Although this work primarily experiments with autoregressive modeling, the tokenizer does not seem to be particularly designed for AR modeling. How would this tokenizer perform on other generative schemes, such as diffusion, and non-autoregressive modeling?

---

> ### Author Response · Authors · 2024-11-17
> **Author response - Part 1**
>
> We thank the reviewer for the time and effort in reviewing our paper.
>
> ---
>
> **"Weakness 1. Some components have been explored before, such as semantic regularization, multi-scale RQ and parallel decoding."**
>
> We respectfully disagree with the claim. There exists some earlier works that share similar concepts, however, completely different in design. Our approach made fundamental improvements compared to previous methods.
>
> - As discussed in Line 295-305, a concurrent work VAIL-U (Arxiv, Sept 24) adopted a semantic constraint targeting understanding capability. However, VALI-U's semantic injection leads to a severe performance drop, i.e., -0.5 rFID and -1.2 gFID. Remarkably, our semantic regularization with an additional branch for detail information leads to a 1.67 rFID and 3.03 gFID gain. We also provided an in-depth analysis discussing the properties of semantic regularization to explain the gain and difference against the previous approach in Line 276-305.
>
> - As discussed in Line 419-426, a baseline approach VAR (NeurIPS 24) proposed multi-scale RQ. However, VAR's token length is significantly large due to the RQ, i.e. 680 tokens. In imagefolder, with the spatial-preserving Q-Former and Product quantization, we successfully reduced the MSVQ's token length to 286 with an even better gFID of 2.6 compared to 3.3 of VAR.
>
> - We did not recognize a work that leverages parallel decoding that is similar to our approach in AR-based image generation. It will be helpful if the reviewer can introduce us to specific papers and we are happy to provide detailed comparison and discussion.
>
> Beyond the above difference, we consider the major contribution of ImageFolder is (1) we explored the tradeoff between rFID and gFID regarding **token length**, and provided a solution to balance them, (2) we analyzed the impact of **semantics** on reconstruction and generation and provided an approach to address both of semantics and details and (3) extensive experiments and discussion towards a **state-of-the-art image tokenizer** (superior rFID and gFID) for generation.
>
> In all, we believe our work provides significant insights and improvements compared to previous methods.
>
> ---
>
> **"Weakness 2. Discussion about product quantization (PQ). More visualization of Fig 8."**
>
> We would like to clarify that we have provided additional visualization similar to Fig 8 in the Appendix (Fig 12). In Lines 203-214, we have provided a detailed formulation of PQ and included sufficient references for readers to understand the concept.
>
> In addition, we would like to clarify that we did not claim tokens have a "disentangled nature" primarily due to PQ (though some audio tokenizer designs, such as Wav2Vec, do have). Instead, we consider the information encoded in each branch to be determined by semantic regularization. As discussed in Line 295-305, one branch is designed to align DINO features. However, since the DINO feature discards high-frequency detail information of images (Table 2), with a reconstruction objective, another branch will capture the complementary detail information. To further facilitate the understanding of PQ, we provided more discussion about the product-quantized space in the revision in the Appendix to enhance the readability.
>
> ---
>
> **"Weakness 3. Missing references, e.g. PQ in audio tokenization."**
>
> We thank the reviewer for introducing us to the related work. We have cited [1] in the revised manuscript.
>
> We would like to further clarify that the mentioned work is a concurrent work published on Arxiv in June 2024, within 4 months of the ICLR ddl according to guidelines. In addition, [1] aims to tackle the low codebook utilization rate in audio tokenization and successfully increase the codebook utilization from 13.5% to 97.5%.
>
> However, the codebook utilization rate of ImageFolder (including all ablations in Tab 3) is 100%. PQ is not the reason why we achieve 100% codebook utilization. Even with just a single branch, we can still achieve 100% utilization. Instead, the introduction of PQ in ImageFolder aims to (1) provide an additional branch for encoding both detail and semantic information, (2) increase the potential combination number in image representation, and (3) shorten the token length.
>
> In this way, we consider our work to have fundamental differences compared to [1].

---

> ### Author Response · Authors · 2024-11-17
> **Author response - Part 2**
>
> **"Question 1. Comparison against other tokenizers. Codebook utilization rate."**
>
> We updated Table 4 to include additional comparisons against other tokenizers. ImageFolder achieves the best performance against previous counterparts in terms of both rFID (0.8) and linear probing accuracy (58.0), including MAGVIT-v2 (0.9 rFID) and VAR's tokenizer (0.9 rFID). Please note that the linear probing is conducted on the quantized tokens instead of intermediate layers in the generator.
>
> | Codebook Size  | 4096 | 8192 | 16384 |
> |-----------------|:-----------------:|:--------------:|:--------------:|
> | Utilization | 100% | 100% | 100% |
> | rFID | 0.80 | 0.70 | 0.67 |
>
> Table A. Codebook utilization of different codebook sizes of ImageFolder tokenizer.
>
> As shown in Table A, we would like to clarify that the codebook utilization rate of ImageFolder is 100% among multiple codebook sizes.
>
> ---
>
> **"Question 2. Performance on other generative schemes."**
>
> We thank the reviewer for raising the generalization request of ImageFolder. A recent exploration of a continuous variant of ImageFolder (without hard quantization) demonstrates remarkable success in diffusion-based models as well.
>
> | Method          | FID$\downarrow$ | IS$\uparrow$ |FID$\downarrow$ (w. cfg)| IS$\uparrow$ (w. cfg) |
> |-----------------|:-----------------:|:--------------:|:-----------------:|:--------------:|
> | SiT-XL [A] | 8.30 | 131.7 | 2.06 | 270.3 |
> | SiT-XL+REPA [B] | 5.90   | 157.8 | 1.80 | 284.0 |
> | **SiT-XL+IC**       | 5.35  | 151.2 | 1.78 | 279.0 |
>
> Table B. Comparison against Diffusion-based models. IC: ImageFolder (continuous variant).
>
> As shown in Table B, we compare ImageFolder (continuous) + SiT-XL [A] against the latest concurrent work REPA [B] + SiT-XL on image generation on ImageNet 256x256 following the diffusion-based convention. We can notice that the ImageFolder can achieve an even better performance than the state-of-the-art diffusion-based approach. This further demonstrates that the ImageFolder's semantic-rich latent representation is vital for image generation regardless of diffusion or autoregressive.
>
> [A] SiT: Exploring Flow and Diffusion-based Generative Models with Scalable Interpolant Transformers, ECCV 2024
>
> [B] Representation Alignment for Generation: Training Diffusion Transformers Is Easier Than You Think, Arxiv Oct 24
>
> ---
>
> Please do not hesitate to let us know if you have any further questions. We hope our response can address your concerns and we would greatly appreciate it if you could increase your rating.

---

> > ### Author Response · Authors · 2024-11-24
> >
> > Dear Reviewer GK7c,
> >
> > As the discussion period draws to a close, we would greatly appreciate your review of our response, and we hope it addresses your concerns. Please do not hesitate to let us know if you have any further questions.
> >
> > Best regards,
> >
> > Authors

---

### Author Response · Authors · 2024-11-17
**General response**

As promised in the original manuscript, we would like to update the final performance of ImageFolder under a more comparable setting, utilizing the DINO discriminator employed in VAR's recipe.

The revised Table 4 now reflects the latest results, showing that ImageFolder achieves the following metrics:

- rFID: **0.80** (compared to VAR: 0.90)
- Linear probing accuracy: **58.0** (compared to LlamaGen: 2.6, VAR: 11.3)
- Token length: **286** (compared to VAR: 680)
- gFID (w. cfg): **2.6**, using ~300M generator parameters in **10** AR steps (compared to VAR: 3.3)
- gFID (w./o. cfg): **6.0** (compared to VAR: 12.0)

---

| Method  | Type | Codebook | Token Length | Utilization Rate$\uparrow$ | rFID$\downarrow$ |
|-----------------|:-----------------:|:--------------:|:--------------:|:--------------:|:--------------:|
| Stable Diffusion-f8 | Continuous | - | 1024 | - | 0.9 |
| Stable Diffusion-f16* | Continuous | - | 256 | - | 1.22 |
| LlamaGen | Discrete | 16384 | 256 | 97% | 2.19 |
| MAGVIT-v2 | Discrete | 262144 | 256 | 100% | ~0.9 |
| VAR | Discrete | 4096 | 680 | 100% | 0.90 |
| **ImageFolder** | Discrete | 4096 * 2 | 286 | 100% | 0.80 |
| **ImageFolder** | Discrete | 8192 * 2 | 286 | 100% | 0.70 |
| **ImageFolder** | Discrete | 16384 * 2 | 286 | 100% | 0.67 |

Table A. Comparison among state-of-the-art image tokenizers. *: reported in MAR (Li et al, 2024) and originally 1.43 FID.

In addition, we would like to compare our tokenizer with other state-of-the-art tokenizers in Table A. We noticed that ImageFolder significantly outperforms discrete counterparts, e.g. MAGVIT-v2 and VAR, and commonly used Stable Diffusion's KL tokenizers. Finally, it is worth mentioning that we will release the ImageFolder code and weights to facilitate the research in image generation.

---

### Meta-Review · Area_Chair_5sJP · 2024-12-22

**Metareview:**

This paper introduces a novel image tokenizer, ImageFolder, which improves the balance between image reconstruction quality and generation quality in visual generative models like diffusion models and autoregressive models. Key contributions include spatially aligned tokens for better generation, product quantization with semantic regularization for enhanced representational capability, and insights into the relationship between token length and quality in image generation.

**Additional Comments On Reviewer Discussion:**

Several concerns were raised by the reviewers, including clarification on the contributions over previous works, details on the adaptation of PQ, etc. After the rebuttal, the majority of the reviewers lean toward accepting the paper. One reviewer who did recommend borderline rejection did not comment on the rebuttal, while the AC read the responses as properly addressing the reviewer's concerns.

---

### Decision · Program_Chairs · 2025-01-22

Accept (Poster)